# Supervised Contrastive Learning for Pre-trained Language Model Fine-tuning

**Beliz Gunel**[†*]**, Jingfei Du**[‡]**, Alexis Conneau**[‡]**, Ves Stoyanov**[‡]
[†]Stanford University, [‡]Facebook AI

## Abstract

State-of-the-art natural language understanding classification models follow two-stages: pre-training a large language model on an auxiliary task, and then fine-tuning the model on a task-specific labeled dataset using cross-entropy loss. However, the cross-entropy loss has several shortcomings that can lead to sub-optimal generalization and instability. Driven by the intuition that good generalization requires capturing the similarity between examples in one class and contrasting them with examples in other classes, we propose a supervised contrastive learning (SCL) objective for the fine-tuning stage. Combined with cross-entropy, our proposed SCL loss obtains significant improvements over a strong RoBERTa-Large baseline on multiple datasets of the GLUE benchmark in few-shot learning settings, without requiring specialized architecture, data augmentations, memory banks, or additional unsupervised data. Our proposed fine-tuning objective leads to models that are more robust to different levels of noise in the fine-tuning training data, and can generalize better to related tasks with limited labeled data.

## 1 Introduction

State-of-the-art for most existing natural language processing (NLP) classification tasks is achieved by models that are first pre-trained on auxiliary language modeling tasks and then fine-tuned on the task of interest with cross-entropy loss (Radford et al., 2019; Howard & Ruder, 2018; Liu et al., 2019; Devlin et al., 2019). Although ubiquitous, the cross-entropy loss – the KL-divergence between one-hot vectors of labels and the distribution of model's output logits – has several shortcomings. Cross entropy loss leads to poor generalization performance (Liu et al., 2016; Cao et al., 2019), and it lacks robustness to noisy labels (Zhang & Sabuncu, 2018; Sukhbaatar et al., 2015) or adversarial examples (Elsayed et al., 2018; Nar et al., 2019). Effective alternatives have been proposed to modify the reference label distributions through label smoothing (Szegedy et al., 2016; Müller et al., 2019), Mixup (Zhang et al., 2018), CutMix (Yun et al., 2019), knowledge distillation (Hinton et al., 2015) or self-training (Yalniz et al., 2019; Xie et al., 2020).

Fine-tuning using cross entropy loss in NLP also tends to be unstable across different runs (Zhang et al., 2020; Dodge et al., 2020), especially when supervised data is limited, a scenario in which pre-training is particularly helpful. To tackle the issue of unstable fine-tuning and poor generalization, recent works propose local smoothness-inducing regularizers (Jiang et al., 2020) and regularization methods inspired by the trust region theory (Aghajanyan et al., 2020) to prevent representation collapse. Empirical evidence suggests that fine-tuning for more iterations, reinitializing top few layers (Zhang et al., 2020), and using debiased Adam optimizer during fine-tuning (Mosbach et al., 2020) can make the fine-tuning stage more stable.

Inspired by the learning strategy that humans utilize when given a few examples, we seek to find the commonalities between the examples of each class and contrast them with examples from other classes. We hypothesize that a similarity-based loss will be able to hone in on the important dimensions of the multidimensional hidden representations hence lead to better few-shot learning results and be more stable while fine-tuning pre-trained language models. We propose a novel objective for fine-tuning that includes a supervised contrastive learning (SCL) term that pushes the examples from the same class close and the examples from different classes further apart. The SCL

---

[*]Work done during Facebook AI research internship, correspondence to `bgunel@stanford.edu`.

term is similar to the contrastive objectives used in self-supervised representation learning across image, speech, and video domains. (Sohn, 2016; Oord et al., 2018; Wu et al., 2018; Bachman et al., 2019; Hénaff et al., 2019; Baevski et al., 2020; Conneau et al., 2020; Tian et al., 2020; Hjelm et al., 2019; Han et al., 2019; He et al., 2020; Misra & Maaten, 2020; Chen et al., 2020a;b). Unlike these methods, however, we use a contrastive objective for supervised learning of the final task, instead of contrasting different augmented views of examples.

In few-shot learning settings (20, 100, 1000 labeled examples), the addition of the SCL term to the fine-tuning objective significantly improves the performance on several natural language understanding classification tasks from the popular GLUE benchmark (Wang et al., 2019) over the very strong baseline of fine-tuning RoBERTa-Large with cross-entropy loss only. Furthermore, pre-trained language models fine-tuned with our proposed objective are not only robust to noise in the fine-tuning training data, but can also exhibit improved generalization to related tasks with limited labeled task data. Our approach does not require any specialized network architectures (Bachman et al., 2019; Hénaff et al., 2019), memory banks (Wu et al., 2018; Tian et al., 2020; Misra & Maaten, 2020), data augmentation of any kind, or additional unsupervised data. To the best of our knowledge, our work is the first to successfully integrate a supervised contrastive learning objective for fine-tuning pre-trained language models. We empirically demonstrate that the new objective has desirable properties across several different settings. Our contributions in this work are listed in the following:

- We propose a novel objective for fine-tuning pre-trained language models that includes a supervised contrastive learning term, as described in Section 2.

- We obtain strong improvements in the few-shot learning settings (20, 100, 1000 labeled examples) as shown in Table 2, leading up to 10.7 points improvement on a subset of GLUE benchmark tasks (SST-2, QNLI, MNLI) for the 20 labeled example few-shot setting, over a very strong baseline – RoBERTa-Large fine-tuned with cross-entropy loss.

- We demonstrate that our proposed fine-tuning objective is more robust, in comparison to RoBERTa-Large fine-tuned with cross-entropy loss, across augmented noisy training datasets (used to fine-tune the models for the task of interest) with varying noise levels as shown in Table 3 – leading up to 7 points improvement on a subset of GLUE benchmark tasks (SST-2, QNLI, MNLI) across augmented noisy training datasets. We use a back-translation model to construct the augmented noisy training datasets of varying noise levels (controlled by the temperature parameter), as described in detail in Section 4.2.

- We show that the task-models fine-tuned with our proposed objective have improved generalizability to related tasks despite having limited availability of labeled task data (Table 7). This led to a 2.9 point improvement on Amazon-2 over the task model fine-tuned with cross-entropy loss only. Moreover, it considerably reduced the variance across few-shot training samples, when transferred from the source SST-2 sentiment analysis task model.

## 2 APPROACH

We propose a novel objective that includes a supervised contrastive learning term for fine-tuning pre-trained language models. The loss is meant to capture the similarities between examples of the same class and contrast them with the examples from other classes.

For a multi-class classification problem with C classes, we work with a batch of training examples of size N, $\{x_i, y_i\}_{i=1,...N}$. $\Phi(\cdot) \in \mathbf{R}^d$ denotes an encoder that outputs the $l_2$ normalized final encoder hidden layer before the softmax projection; $N_{y_i}$ is the total number of examples in the batch that have the same label as $y_i$; $\tau > 0$ is an adjustable scalar temperature parameter that controls the separation of classes; $y_{i,c}$ denotes the label and $\hat{y}_{i,c}$ denotes the model output for the probability of the ith example belonging to the class c; $\lambda$ is a scalar weighting hyperparameter that we tune for each downstream task and setting. The overall loss is then given in the following:

$$\mathcal{L} = (1 - \lambda)\mathcal{L}_{CE} + \lambda\mathcal{L}_{SCL} \quad (1)$$

$$\mathcal{L}_{CE} = -\frac{1}{N}\sum_{i=1}^{N}\sum_{c=1}^{C} y_{i,c} \cdot log\hat{y}_{i,c} \quad (2)$$

$$\mathcal{L}_{SCL} = \sum_{i=1}^{N} -\frac{1}{N_{y_i} - 1}\sum_{j=1}^{N} \mathbf{1}_{i \neq j}\mathbf{1}_{y_i = y_j} \log \frac{\exp\left(\Phi(x_i) \cdot \Phi(x_j)/\tau\right)}{\sum_{k=1}^{N}\mathbf{1}_{i \neq k}\exp\left(\Phi(x_i) \cdot \Phi(x_k)/\tau\right)} \quad (3)$$

The overall loss is a weighted average of CE and the proposed SCL loss, as given in the equation (1). The canonical definition of the multi-class CE loss that we use is given in equation (2). The novel SCL loss is given in the equation (3).

This loss can be applied using a variety of encoders $\Phi(\cdot) \in \mathbf{R}^d$ – for example a ResNet for a computer vision application or a pre-trained language model such as BERT for an NLP application. In this work, we focus on fine-tuning pre-trained language models for single sentence and sentence-pair classification settings. For single sentence classification, each example $x_i$ consists of sequence of tokens prepended with the special $[CLS]$ token $x_i = [[CLS], t_1, t_2, \ldots, t_L, [EOS]]$. The length of sequence L is constrained such that $L < L_{\max}$. Similarly, for sentence-pair classification tasks, each example $x_i$ is a concatenation of two sequences of tokens $[t_1, t_2, \ldots t_L]$ and $[s_1, s_2, \ldots, s_M]$ corresponding to the sentences with special tokens delimiting them: $x_i = [[CLS], t_1, t_2, \ldots, t_L, [SEP], s_1, s_2, \ldots, s_M, [EOS]]$. The length of concatenated sequences is constrained such that $L + M < L_{\max}$. In both cases, $\Phi(x_i) \in \mathbf{R}^d$ uses the embedding of $[CLS]$ token as the representation for example $x_i$. These choices follow standard practices for fine-tuning pre-trained language models for classification (Devlin et al., 2019; Liu et al., 2019).

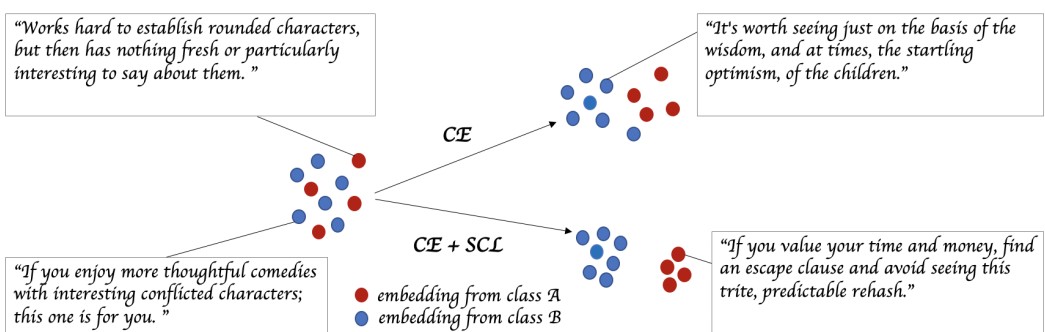

Figure 1: Our proposed objective includes a cross-entropy term (CE) and a supervised contrastive learning (SCL) term, and it is formulated to push examples from the same class close and examples from different classes further apart. We show examples from the SST-2 sentiment analysis dataset from the GLUE benchmark, where class A (shown in red) is negative movie reviews and class B (shown in blue) is positive movie reviews. Although we show a binary classification case for simplicity, the loss is generally applicable to any multi-class classification setting.

Empirical observations show that both $l_2$ normalization of the encoded embedding representations and an adjustable scalar temperature parameter $\tau$ improve performance. Lower temperature increases the influence of examples that are harder to separate, effectively creating harder negatives. Using hard negatives has been previously shown to improve performance in the context of margin-based loss formulations such as triplet loss (Schroff et al., 2015). The empirical behavior of the adjustable temperature parameter is consistent with the observations of previous work related to supervised contrastive learning. (Chen et al., 2020a; Khosla et al., 2020).

**Relationship to Self-Supervised Contrastive Learning** Self-supervised contrastive learning has shown success in learning powerful representations, particularly in the computer vision domain. (Chen et al., 2020a; He et al., 2020; Tian et al., 2020; Mnih & Kavukcuoglu, 2013; Gutmann & Hyvärinen, 2012; Kolesnikov et al., 2019) Self-supervised learning methods do not require any labeled data; instead they sample a mini batch from unsupervised data and create *positive* and *negative* examples

from these samples using strong data augmentation techniques such as AutoAugment (Cubuk et al., 2019) or RandAugment (Cubuk et al., 2020) for computer vision. *Positive* examples are constructed by applying data augmentation to the same example (cropping, flipping, etc. for an image), and *negative* examples are simply all the other examples in the sampled mini batch. Intuitively, self-supervised contrastive objectives are learning representations that are invariant to different views of *positive* pairs; while maximizing the distance between *negative* pairs. The distance metric used is often the inner product or the Euclidean distance between vector representations of the examples.

For a batch of size N, self-supervised contrastive loss is defined as:

$$\mathcal{L}_{self} = \sum_{i=1}^{2N} - \log \frac{\exp\left(\Phi(x'_{2i-1}) \cdot \Phi(x'_{2i})/\tau\right)}{\sum_{k=1}^{2N} \mathbf{1}_{i \neq k} \exp\left(\Phi(x'_i) \cdot \Phi(x'_k)/\tau\right)} \tag{4}$$

where $\Phi(\cdot) \in \mathbf{R}^d$ denotes an encoder that outputs the $l_2$ normalized final encoder hidden layer before the softmax projection; $\tau > 0$ is a scalar temperature parameter. $\mathbf{A}$ is defined as a data augmentation block that generates two randomly generated augmented examples, $x'_{2i}$ and $x'_{2i-1}$ from the original example $x_i$: $\mathbf{A}(\{x_i, y_i\}_{i=1,\dots N}) = \{x'_i, y'_i\}_{i=1,\dots 2N}$. As an example, $\mathbf{A}$ can be RandAugment for a computer vision application; or it could be a back-translation model for an NLP application.

## 3 RELATED WORK

**Traditional Machine Learning and Theoretical Understanding** Several works have analyzed the shortcomings of the widely adopted cross-entropy loss, demonstrating that it leads to poor generalization performance due to poor margins (Liu et al., 2016; Cao et al., 2019), and lack of robustness to noisy labels (Zhang & Sabuncu, 2018; Sukhbaatar et al., 2015) or adversarial examples (Elsayed et al., 2018; Nar et al., 2019). On the other hand, there has been a body of work that has explored the performance difference for classifiers trained with discriminative (i.e., optimizing for $p(y|x)$, where y is the label and x is the input) losses such as cross-entropy loss and generative losses (i.e. optimizing for $p(x|y)$). Ng & Jordan (2001) show that classifiers trained with generative losses can outperform their counterparts trained with discriminative losses in the context of Logistic Regression and Naive Bayes. Raina et al. (2003) show that a hybrid discriminative and generative objective outperforms both solely discriminative and generative approaches. In the context of contrastive learning, Saunshi et al. (2019) propose a theoretical framework for analyzing contrastive learning algorithms through hypothesizing that semantically similar points are sampled from the same latent class, which allows showing formal guarantees on the quality of learned representations.

**Contrastive Learning** There has been several recent investigations for the use of contrastive objectives for self-supervised, semi-supervised, and supervised learning methods, primarily in the computer vision domain. Chen et al. (2020a) propose a framework for contrastive learning of visual representations without specialized architectures or a memory bank, and show state-of-the-art results on ImageNet ILSVRC-2012 (Russakovsky et al., 2015) – outperforming previous methods for self-supervised, semi-supervised and transfer learning. Similarly, Khosla et al. (2020) propose a supervised contrastive loss that outperforms cross entropy loss and gets state-of-the-art results on ImageNet on both ResNet-50 and ResNet-200 (He et al., 2016) with AutoAugment (Cubuk et al., 2019) data augmentation. They also show increased robustness on the ImageNet-C dataset (Hendrycks & Dietterich, 2019), and demonstrate that supervised contrastive loss is less sensitive to different hyperparameter settings for optimizers or data augmentations compared to the cross-entropy loss. Liu & Abbeel (2020) propose a hybrid discriminative-generative training of energy-based models where they approximate the generative term with a contrastive loss using large batch sizes and show improved classification accuracy of WideResNet-28-10 (Zagoruyko & Komodakis, 2016) on CIFAR-10 and CIFAR-100 (Krizhevsky, 2009) datasets, outperforming state-of-the-art discriminative and generative classifiers. They also demonstrate improved performance for WideResNet-28-10 on robustness, out-of-distribution detection, and calibration, compared to other state-of-the-art generative and hybrid models. Finally, Fang & Xie (2020) propose pre-training language models using a self-supervised contrastive learning objective at the sentence level using back-translation as the augmentation method, followed by fine-tuning by predicting whether two augmented sentences originate from the same sentence – demonstrating improvements over fine-tuning BERT on a subset of GLUE benchmark tasks.

**Stability and Robustness of Fine-tuning Pre-trained Language Models** There has been recent works on analyzing the stability and robustness of fine-tuning pre-trained language models, since they have been shown to overfit to the labeled task data while fine-tuning and hence fail to generalize to unseen data when there is limited labeled data for the task (Aghajanyan et al., 2020). To improve the generalization performance, Jiang et al. (2020) propose a local smoothness-inducing regularizer to manage the complexity of the model and a Bregman proximal point optimization method, an instance of trust-region methods, to prevent aggressive updating of the model during fine-tuning. They show state-of-the-art performance on GLUE, SNLI (Bowman et al., 2015), SciTail (Khot et al., 2018), and ANLI (Nie et al., 2020) natural language understanding benchmarks. Similarly, Aghajanyan et al. (2020) propose a regularized fine-tuning procedure inspired by trust-region theory that replaces adversarial objectives with parametric noise sampled from normal or uniform distribution in order to prevent representation collapse during fine-tuning for better generalization performance, without hurting the performance. They show improved performance on a range of natural language understanding and generation tasks including DailyMail/CNN (Hermann et al., 2015), Gigaword (Napoles et al., 2012), Reddit TIFU (Kim et al., 2019), and the GLUE benchmark. There has also been some empirical analysis that suggests fine-tuning for more epochs, reinitializing top few layers (Zhang et al., 2020) instead of only the classification head, and using debiased Adam optimizer instead of BERTAdam (Devlin et al., 2019) during fine-tuning (Mosbach et al., 2020) can make the fine-tuning procedure more stable across different runs.

## 4 EXPERIMENTAL SETUP

### 4.1 DATASETS AND TRAINING DETAILS

We use datasets from the GLUE natural language understanding benchmark (Wang et al., 2019) for evaluation. We include both single sentence classification tasks and sentence-pair classification tasks to test whether our hypothesis is generally applicable across tasks. We summarize each dataset based on their main task, domain, number of training examples, and number of classes in Table 1.

In our few-shot learning experiments, we sample half of the original validation set of the GLUE benchmark and use it as our test set, and sample ∼500 examples for our validation set from the original GLUE validation set, both taking the label distribution of the original validation set into account. For each task, we want the validation set to be small enough to avoid easy overfitting on the validation set, and big enough to avoid high-variance when early-stopping at various epochs for the few-shot learning experiments. For full dataset experiments, such as the ones shown in Table 5, Table 6, Table 8, and Table 9, we sample a validation set from the original training set of the GLUE benchmark based on the size of the original validation set of GLUE, and report our test results on the original validation set of GLUE.

We run each experiment with 10 different seeds, and report the average test accuracy, standard deviation, along with p-values with respect to the baseline. We pick the best hyperparameter combination based on the average validation accuracy across 10 seeds. For few-shot learning experiments, such as the ones shown in Table 2, Table 3, and Table 10, we sample 10 different training set samples based on the total number of examples $N$ specified from the original training set of the GLUE benchmark, taking the label distribution of the original training set into account. We report the average and the standard deviation of the test accuracies of the top 3 models based on their validation accuracies out of 10 random training set samples. Best hyperparameter combination is picked based on the average validation accuracy of the top 3 models. The reason why we focus on the top 3 models for this setting is that we would like to reduce the variance across training set samples.

We use fairseq Ott et al. (2019) library and the open-source RoBERTa-Large model for all of our experiments. During all the fine-tuning runs, we use Adam optimizer with a learning rate of 1e-5, batch size of 16 (unless specified otherwise), and dropout rate of 0.1. For each experiment that includes the SCL term, we conduct a grid-based hyperparameter sweep for $\lambda \in \{0.1, 0.3, 0.5, 0.7, 0.9, 1.0\}$ and $\tau \in \{0.1, 0.3, 0.5, 0.7\}$. We observe that models with best test accuracies across all experimental settings overwhelmingly use the hyperparameter combination $\tau = 0.3$ and $\lambda = 0.9$.

| Dataset | Task | Domain | #Train | #Classes |
|---------|------|--------|--------|----------|
| SST-2 | sentiment analysis | movie reviews | 67k | 2 |
| CoLA | grammatical correctness | linguistic publications | 8.5k | 2 |
| MRPC | paraphrase | news | 3.7k | 2 |
| RTE | textual entailment | news/Wikipedia | 2.5k | 2 |
| QNLI | question answering/textual entailment | Wikipedia | 105k | 2 |
| MNLI | textual entailment | multi-domain | 393k | 3 |

Table 1: GLUE Benchmark datasets used for evaluation.

## 4.2 CONSTRUCTING AUGMENTED NOISY TRAINING DATASETS

Machine learning researchers or practitioners often do not know how noisy their datasets are, as input examples might be corrupted or ground truth labeling might not be perfect. Therefore, it is preferable to use robust training objectives that can get more information out of datasets of different noise levels, even where there is limited amount of labeled data. We construct augmented noisy training datasets (used to fine-tune the pre-trained language models for the task of interest) of different noise levels using a back-translation model (Edunov et al., 2018), where we increase the temperature parameter to create more noisy examples. Back-translation refers to the procedure of translating an example in language A into language B and then translating it back to language A, and it is a commonly used data augmentation procedure for NLP applications, as the new examples obtained through back-translation provide targeted inductive bias to the model while preserving the meaning of the original example. Specifically, we use WMT'18 English-German and German-English translation models, use random sampling to get more diverse examples, and employ and augmentation ratio of 1:3 for supervised examples:augmented examples. We observe that employing random sampling with a tunable temperature parameter is critical to get diverse paraphrases for the supervised examples, consistent with the previous work (Edunov et al., 2018; Xie et al., 2019), since commonly used beam search results in very regular sentences that do not provide diversity to the existing data distribution. We keep the validation and test sets same with the experiments shown in Table 2.

## 5 ANALYSIS AND RESULTS

### 5.1 GLUE BENCHMARK FEW-SHOT LEARNING RESULTS

We proposed adding the SCL term inspired by the learning strategy of humans when they are given few examples. In Table 2, we report our few-shot learning results on SST-2, QNLI, and MNLI from the GLUE benchmark with 20, 100, 1000 labeled training examples. Details of the experimental setup are explained in Section 4. We use a very strong baseline of fine-tuning RoBERTa-Large with cross-entropy loss. We observe that the SCL term improves performance over the baseline significantly across all datasets and data regimes, leading to 10.7 points improvement on QNLI, 3.4 points improvement on MNLI, and 2.2 points improvement on SST-2, where we have 20 labeled examples for fine-tuning. This shows that our proposed objective is effective both for binary single sentence classification such as sentiment analysis; and sentence pair classification tasks such as textual entailment and paraphrasing – when we are given only few labeled examples for the task. We see that as we increase the number of labeled examples, performance improvement over the baseline decreases, leading to 1.9 points improvement on MNLI for 100 examples and 0.6 points improvement on QNLI for 1000 examples. We also would like to acknowledge that improvements over the baseline when N=1000 on both SST-2 and MNLI are not statistically significant. In addition, we conduct an ablation study where we investigate the importance of $l_2$ normalization and temperature scaling where we replace SCL loss with CE loss but keep the $l_2$ normalization and temperature scaling, as shown in Table 10 in the Appendix under the method name CE+CE.

In Figure 2, we show tSNE plots of the learned representations of the CLS embeddings on SST-2 test set when RoBERTa-Large is fine-tuned with 20 labeled examples, comparing CE with and without the SCL term. We can clearly see that the SCL term enforces more compact clustering of examples with the same label; while the distribution of the embeddings learned with CE is close to random. We include a more detailed comparison for CE and CE+SCL showing learned representations of

examples as tSNE plots, where we have 20, 100 labeled examples and full dataset respectively for fine-tuning in Figure 3 in the Appendix.

| Model | Loss | N | SST-2 | QNLI | MNLI |
|---|---|---|---|---|---|
| RoBERTa$_{Large}$ | CE | 20 | 85.9±2.1 | 65.0±2.0 | 39.3±2.5 |
| RoBERTa$_{Large}$ | CE + SCL | 20 | **88.1±3.3** | **75.7±4.8** | **42.7±4.6** |
| | p-value | | 5e-10 | 1e-46 | 1e-8 |
| RoBERTa$_{Large}$ | CE | 100 | 91.1±1.3 | 81.9±0.4 | 59.2±2.1 |
| RoBERTa$_{Large}$ | CE + SCL | 100 | **92.8±1.3** | **82.5±0.4** | **61.1±3.0** |
| | p-value | | 3e-17 | 1e-20 | 2e-4 |
| RoBERTa$_{Large}$ | CE | 1000 | 94.0±0.6 | 89.2±0.6 | 81.4±0.2 |
| RoBERTa$_{Large}$ | CE + SCL | 1000 | **94.1±0.5** | **89.8±0.4** | **81.5±0.2** |
| | p-value | | 0.6 | 1e-12 | 0.5 |

Table 2: Few-shot learning test results on the GLUE benchmark where we have N=20,100,1000 labeled examples for training. Reported results are the mean and the standard deviation of the test accuracies of the top 3 models based on validation accuracy out of 10 random training set samples, along with p-values for each experiment.

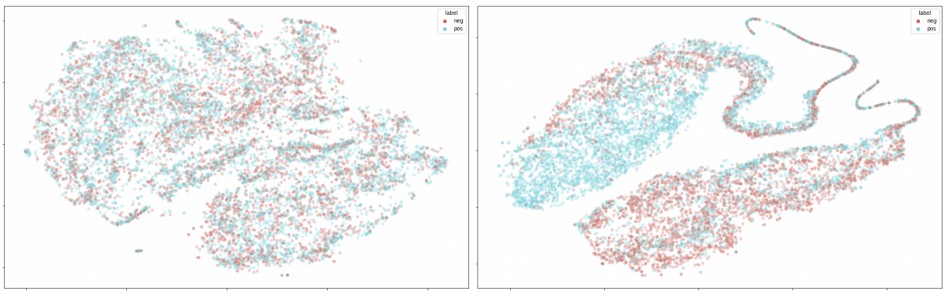

Figure 2: tSNE plots of the learned CLS embeddings on the SST-2 test set in the few-shot learning setting of having 20 labeled examples to fine-tune on – comparing RoBERTa-Large fine-tuned with CE only (left) and with our proposed objective CE+SCL (right) for the SST-2 sentiment analysis task. Blue: positive examples; red: negative examples.

## 5.2 ROBUSTNESS ACROSS AUGMENTED NOISY TRAINING DATASETS

In Table 3, we report our results on augmented noisy training sets with varying levels of noise. We have 100 labeled examples for fine-tuning for each task, and we augment their training sets with noisy examples using a back-translation model, as described in detail in Section 4.2. Note that we use the back-translation model to simulate training datasets of varying noise levels and not as a method to boost model performance. Experimental setup follows what is described in Section 4 for few-shot learning experiments. T is the temperature for the back-translation model used to augment the training sets, and higher temperature corresponds to more noise in the augmented training set.

We observe consistent improvements over the RoBERTa-Large baseline with our proposed objective across all datasets across all noise levels, with 0.4 points improvement on SST-2, 2.5 points improvement on QNLI, and 7 points improvement on MNLI on average across augmented training sets. The improvement is particularly significant for inference tasks (QNLI, MNLI) when the noise levels are higher (higher temperature), leading to 7.7 points improvement on MNLI when T=0.7, and 4.2 points improvement on QNLI when T=0.9. We show some samples of the augmented examples used in this robustness experiment in Table 4. For T=0.3, examples mostly stay the same with minor changes in their phrasing, while for T=0.9, some grammatical mistakes and factual errors are introduced.

| Dataset | Loss | Original | T=0.3 | T=0.5 | T=0.7 | T=0.9 | Average |
|---------|------|----------|-------|-------|-------|-------|---------|
| SST-2 | CE | 91.1±1.3 | 92.0±1.3 | 91.4±1.0 | **91.7±1.3** | 90.0±0.5 | 91.3±1.2 |
| SST-2 | CE + SCL | **92.8±1.3** | **92.6±0.9** | **91.5±1.0** | 91.2±0.6 | **91.5±1.0** | **91.7±1.0** |
| QNLI | CE | 81.9±0.4 | 81.1±2.3 | 80.0±2.9 | 78.9±3.7 | 75.9±4.0 | 79.0±3.5 |
| QNLI | CE + SCL | **82.5±0.4** | **82.7±1.9** | **81.9±2.5** | **81.3±0.6** | **80.1±2.5** | **81.5±2.0** |
| MNLI | CE | 59.2±2.1 | 54.0±1.1 | 55.3±2.4 | 54.6±2.2 | 47.0±1.8 | 52.7±3.9 |
| MNLI | CE + SCL | **61.1±3.0** | **61.2±2.3** | **62.1±0.9** | **62.3±1.1** | **53.0±2.1** | **59.7±4.3** |

Table 3: Results on the GLUE benchmark for robustness across noisy augmented training sets. Average shows the average performance across augmented training sets.

| Dataset | Type | Sentence |
|---------|------|----------|
| SST-2 | Original | As possibly the best actor working in movies today. |
| SST-2 | Augmented (T=0.3) | As perhaps the best actor who now stars in films. |
| SST-2 | Original | The young stars are too cute; the story and ensuing complications are too manipulative. |
| SST-2 | Augmented (T=0.9) | The babies are too cute, the image and complications that follow too manipulative. |
| QNLI | Original | Brain tissue is naturally soft, but can be stiffened with what liquid? |
| QNLI | Augmented (T=0.3) | Brain tissue is omitted naturally, but with what fluid it can be stiffened? |
| QNLI | Original | In March 1968, CBS and Sony formed CBS/Sony Records, a Japanese business joint venture. |
| QNLI | Augmented (T=0.9) | CBS was founded by CBS and Sony Records in March 1962, a Japanese company. |
| MNLI | Original | However, the link did not transfer the user to a comment box particular to the rule at issue. |
| MNLI | Augmented (T=0.3) | However, the link did not send the user to a comment field specifically for the rule. |
| MNLI | Original | Tenants could not enter the apartment complex due to a dangerous chemical spill. |
| MNLI | Augmented (T=0.9) | Tenants were banned from entering the medical property because of a blood positive substance. |

Table 4: Sample of augmented examples with different noise levels for the robustness experiment shown in Table 3. Higher temperature (T) corresponds to more noise in the augmented training set.

## 5.3 GLUE BENCHMARK FULL DATASET RESULTS

In Table 5, we report results using our proposed objective on six downstream tasks from the GLUE benchmark. We use a very strong baseline of fine-tuning RoBERTa-Large with cross-entropy loss, which is currently the standard practice for the state-of-the-art NLP classification models. Details of the experimental setup are explained in Section 4.

We observe that adding the SCL term to the objective improves the performance over the RoBERTa-Large baseline that lead to 3.1 points improvement on MRPC, 3.5 points improvement on QNLI, and an average improvement of 1.2 points across all 6 datasets. We conduct these experiments to investigate the effect of the SCL term in high-data regimes, as we observe that it's effective in few-shot learning settings. We acknowledge that only MRPC and QNLI results are statistically significant, and we report the results on the other datasets as a finding for the sake of completeness.

We hypothesize larger batch sizes lead to better performance, but we leave that for future work as that requires additional engineering effort. We show evidence for this hypothesis in our ablation studies that we show in Table 6, where we conduct the full dataset experiments for CE+SCL with the same experimental setup described here for Table 5 on SST-2, CoLA, QNLI, and MNLI for batch sizes 16, 64, and 256 using RoBERTa-Base. We observe that as we increase the batch size, performance improves significantly across all datasets. Specifically, we observe 0.3 points improvement on SST-2, 0.8 points improvement on CoLA, 0.4 points improvement on QNLI, and 1.3 points improvement on MNLI, when we increase the batch size from 16 to 256 for CE+SCL. We also investigate the effect of SCL term in the overall training speed, and we measure that with average updates per second metric, shown in Table 6. For batch size 16, the batch size we use throughout the paper across all experimental settings, effect of SCL is negligible – decreasing average updates per second from 15.9 to 15.08. As we increase the batch size, effect of SCL to training speed becomes more significant – decreasing average updates per second from 2.46 to 1.54 for batch size 256. In addition, we conduct an ablation study where we investigate the importance of $l_2$ normalization and temperature scaling where we replace SCL loss with CE loss but keep the normalization and scaling (denoted as CE+CE) both for full dataset results in Table 8, and for batch size ablation in Table 9 in the Appendix.

| Model | Loss | SST-2 | CoLA | MRPC | RTE | QNLI | MNLI | Avg |
|---|---|---|---|---|---|---|---|---|
| RoBERTa$_{Large}$ | CE | 96.0±0.4 | 86.0±0.5 | 86.4±2.4 | 85.5±1.8 | 90.4±0.8 | 88.4±1 | 88.8 |
| RoBERTa$_{Large}$ | CE + SCL | **96.3±0.4** | **86.1±0.8** | **89.5±0.9** | **85.7±0.5** | **93.9±0.7** | **88.6±0.7** | **90** |
| | p-value | 0.07 | 0.63 | 0.01 | 0.06 | 0.01 | 0.16 | |

Table 5: Test results on the validation set of GLUE benchmark. We compare fine-tuning RoBERTa-Large with CE with and without SCL. Best hyperparameter configuration picked based on average validation accuracy. We report average accuracy across 10 seeds for the model with best hyperparameter configuration, its standard deviation, and p-values.

| Model | Loss | Bsz | SST-2 | CoLA | QNLI | MNLI | Avg ups/sec |
|---|---|---|---|---|---|---|---|
| RoBERTa$_{Base}$ | CE | 16 | 94.1±0.5 | 83.3±0.7 | 88.2±0.8 | 84±0.6 | 15.9 |
| RoBERTa$_{Base}$ | CE + SCL | 16 | 94.9±0.6 | 83.7±0.9 | 92.5±0.4 | 85.3±0.5 | 15.08 |
| RoBERTa$_{Base}$ | CE | 64 | 94.2±0.4 | 83.3±0.5 | 89.2±0.5 | 84±0.4 | 8.43 |
| RoBERTa$_{Base}$ | CE + SCL | 64 | 94.7±0.2 | 83.8±0.6 | 92.6±0.5 | 85.7±0.7 | 7.44 |
| RoBERTa$_{Base}$ | CE | 256 | 94.1±0.4 | 84±0.5 | 90±0.7 | 84.4±0.6 | **2.46** |
| RoBERTa$_{Base}$ | CE + SCL | 256 | **95.2±0.3** | **84.5±0.5** | **92.9±0.3** | **86.6±0.6** | 1.54 |

Table 6: Ablation study on performance and training speed shown as average updates per second (Avg ups/sec) for fine-tuning RoBERTa-Base with respect to the batch size (Bsz).

## 5.4 GENERALIZATION ABILITY OF TASK MODELS

In this experiment, we first fine-tune RoBERTa-Large on SST-2 using its full training set and get a task model with and without SCL term. Then, we transfer this task model to two related single sentence sentiment analysis binary classification tasks for the movie reviews domain – Amazon-2 and Yelp-2 (Zhang et al., 2015). For both, we sample 20 labeled examples for each class, and follow the few-shot learning experimental setup described in Section 4. In Table 7, we demonstrate that using the SCL term for both source (SST-2) and target domains (Amazon-2, Yelp-2) lead to better generalization ability, with 2.9 points improvement on Amazon-2 and 0.4 points improvement on Yelp-2 along with significant reduction in variance across training set samples.

| Model | Loss | N | Amazon-2 | Yelp-2 |
|---|---|---|---|---|
| RoBERTa$_{Large}$ | CE | 40 | 87.4±6.4 | 90.8±2.2 |
| RoBERTa$_{Large}$ | CE + SCL | 40 | **90.3±0.6** | **91.2±0.4** |

Table 7: Generalization of the SST-2 task model (fine-tuned using the full training set) to related tasks (Amazon-2, Yelp-2) where there are 20 labeled examples for each class.

## 6 CONCLUSION

We propose a supervised contrastive learning objective for fine-tuning pre-trained language models and demonstrate significant improvements over a strong RoBERTa-Large baseline on multiple datasets of the GLUE benchmark in the few-shot learning settings. We also show that our proposed objective leads to models that are more robust to different levels of noise in the training data and can generalize better to related tasks with limited labeled task data. Currently, data augmentation methods in NLP and their effects on the downstream tasks are neither as effective nor as well understood as their counterparts in the computer vision domain. In future work, we plan to study principled and automated data augmentation techniques for NLP that would allow extending our supervised contrastive learning objective to both semi-supervised and self-supervised learning settings.

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

# A  APPENDIX

*# labeled examples*

*full*

*100*

*20*

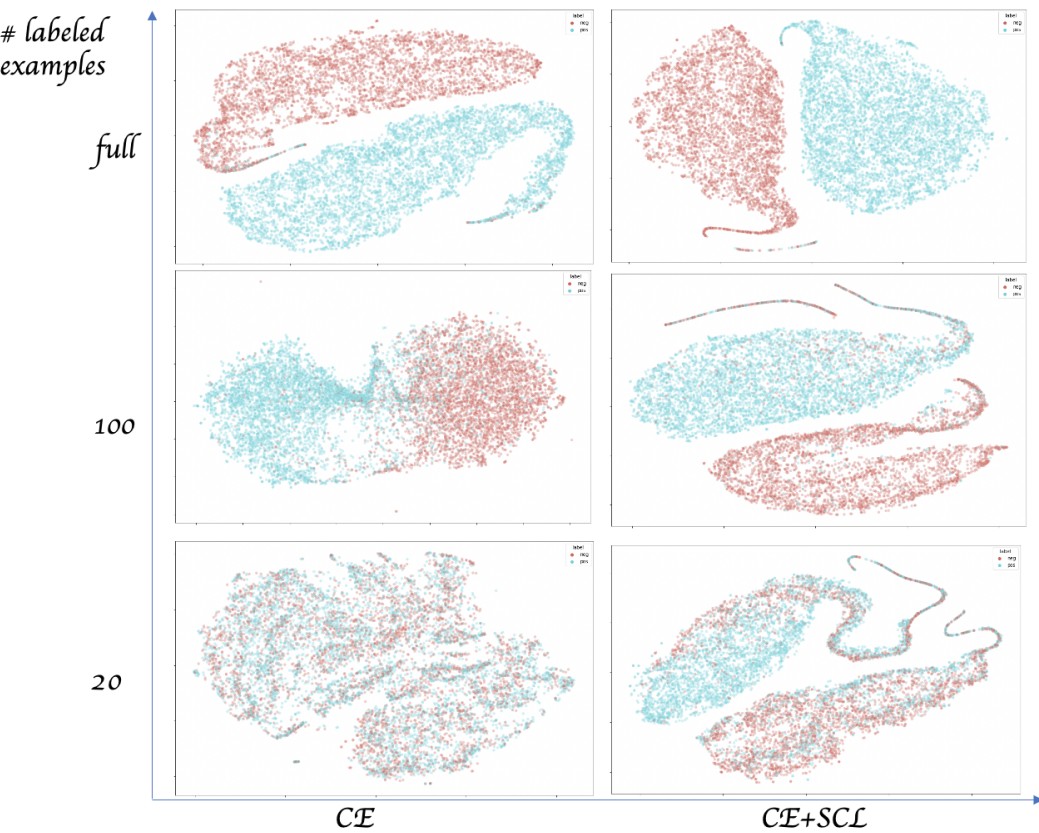

*CE*                   *CE+SCL*

Figure 3: tSNE plots of learned CLS embedding on SST-2 test set where we have 20, 100 labeled examples, and full dataset respectively, comparing CE with and without SCL term. Blue: positive examples; red: negative examples.

| Model | Loss | SST-2 | CoLA | MRPC | RTE | QNLI | MNLI | Avg |
|---|---|---|---|---|---|---|---|---|
| RoBERTa$_{\text{Large}}$ | CE | 96.0±0.4 | 86.0±0.5 | 86.4±2.4 | 85.5±1.8 | 90.4±0.8 | 88.4±1 | 88.8 |
| RoBERTa$_{\text{Large}}$ | CE + SCL | **96.3±0.4** | 86.1±0.8 | **89.5±0.9** | **85.7±0.5** | **93.9±0.7** | 88.6±0.7 | **90** |
| | p-value | 0.07 | 0.63 | 0.01 | 0.06 | 0.01 | 0.16 | |
| RoBERTa$_{\text{Large}}$ | CE + CE | 96±0.4 | 86.3±0.4 | 89±1 | 84.9±1 | 93.9±0.8 | **89±1** | 89.9 |
| | p-value | 0.39 | 0.13 | 0.01 | 0.1 | 0.01 | 0.12 | |
| RoBERTa$_{\text{Large}}$ | Khosla et al. (2020) | 96±0.3 | **86.7±1** | 89.3±1.2 | 85.2±1 | 92.4±0.7 | 88.8±0.9 | 89.7 |
| | p-value | 0.4 | 0.42 | 0.01 | 0.22 | 0.01 | 0.13 | |

Table 8: Test results on the validation set of GLUE benchmark. We compare fine-tuning RoBERTa-Large with CE with and without SCL, CE+CE and the two-stage method of Khosla et al. (2020). Best hyperparameter configuration is picked based on the average validation accuracy. We report average accuracy across 10 seeds for the model with the best hyperparameter configuration, its standard deviation, and p-values. CE+CE refers to the case where we replace SCL loss with the CE loss but keep l2 normalization and temperature scaling.

| Model | Loss | Bsz | SST-2 | CoLA | QNLI | MNLI | Avg ups/sec |
|---|---|---|---|---|---|---|---|
| RoBERTa$_{Base}$ | CE | 16 | 94.1±0.5 | 83.3±0.7 | 88.2±0.8 | 84±0.6 | 15.9 |
| RoBERTa$_{Base}$ | CE + SCL | 16 | 94.9±0.6 | 83.7±0.9 | 92.5±0.4 | 85.3±0.5 | 15.08 |
| RoBERTa$_{Base}$ | CE + CE | 16 | 94.8±0.7 | 83.6±0.4 | 91.6±0.5 | 85±0.3 | 15.25 |
| RoBERTa$_{Base}$ | CE | 64 | 94.2±0.4 | 83.3±0.5 | 89.2±0.5 | 84±0.4 | 8.43 |
| RoBERTa$_{Base}$ | CE + SCL | 64 | 94.7±0.2 | 83.8±0.6 | 92.6±0.5 | 85.7±0.7 | 7.44 |
| RoBERTa$_{Base}$ | CE + CE | 64 | 94.6±0.7 | 83.5±0.6 | 92.1±0.8 | 85±0.8 | 7.64 |
| RoBERTa$_{Base}$ | CE | 256 | 94.1±0.4 | 84±0.5 | 90±0.7 | 84.4±0.6 | **2.46** |
| RoBERTa$_{Base}$ | CE + SCL | 256 | **95.2±0.3** | **84.5±0.5** | **92.9±0.3** | **86.6±0.6** | 1.54 |
| RoBERTa$_{Base}$ | CE + CE | 256 | 94.3±0.5 | 83.5±0.3 | 91.9±0.4 | 84.6±0.8 | 1.77 |

Table 9: Ablation on performance and fine-tuning speed shown as average updates per second (Avg ups/sec) for fine-tuning RoBERTa-Base with respect to the batch size (Bsz). CE+CE refers to the case where we replace SCL loss with the CE loss but keep l2 normalization and temperature scaling.

| Model | Loss | N | SST-2 | QNLI | MNLI |
|---|---|---|---|---|---|
| RoBERTa$_{Large}$ | CE | 20 | 85.9±2.1 | 65.0±2.0 | 39.3±2.5 |
| RoBERTa$_{Large}$ | CE + SCL
p-value | 20 | **88.1±3.3**
5e-10 | **75.7±4.8**
1e-46 | **42.7±4.6**
1e-8 |
| RoBERTa$_{Large}$ | CE + CE
p-value | 20 | 86.5±2.2
0.03 | 75.1±3.5
4e-68 | 40.8±3.7
3e-4 |
| RoBERTa$_{Large}$ | CE | 100 | 91.1±1.3 | 81.9±0.4 | 59.2±2.1 |
| RoBERTa$_{Large}$ | CE + SCL
p-value | 100 | **92.8±1.3**
3e-17 | **82.5±0.4**
1e-20 | **61.1±3.0**
2e-4 |
| RoBERTa$_{Large}$ | CE + CE
p-value | 100 | 91.7±0.5
1e-4 | 81.7±0.5
3e-4 | 56±4.0
2e-8 |
| RoBERTa$_{Large}$ | CE | 1000 | 94.0±0.6 | 89.2±0.6 | 81.4±0.2 |
| RoBERTa$_{Large}$ | CE + SCL
p-value | 1000 | **94.1±0.5**
0.6 | **89.8±0.4**
1e-12 | **81.5±0.2**
0.5 |
| RoBERTa$_{Large}$ | CE + CE
p-value | 1000 | 94±0.7
0.78 | 89.3±1
0.06 | 81.2±0.2
0.12 |

Table 10: Few-shot learning test results on the GLUE benchmark where we have N=20,100,1000 labeled examples for fine-tuning. Reported results are the mean and the standard deviation of the test accuracies of the top 3 models based on the validation accuracy out of 10 random training set samples, along with p-values for each experiment. CE+CE refers to the case where we replace SCL loss with the CE loss but keep l2 normalization and temperature scaling.

