# OpenReview forum: "Supervised Contrastive Learning for Pre-trained Language Model Fine-tuning"
_ICLR.cc/2021/Conference — ICLR 2021 Poster_

### Official Review · AnonReviewer2 · 2020-10-25
**Clear goal, clear presentation, somewhat limited novelty / impact**

**Rating:** 6
**Confidence:** 4

**Review:**

The paper proposes a new training objective for fine-tuning pre-trained models: a weighted sum of the classical cross-entropy (CE) and a new supervised contrastive learning term (SCP). The latter uses the (negated) softmax over the embedding distances (i.e. dot products) between a training instance and all other instances in the batch with the same label. In contrast to the more traditional self-supervised contrastive learning (where positive pairs are obtained by applying transformations to the original data instance), there is no data augmentation; two examples with the same label constitute a positive pair.

Experiments on the GLUE benchmark compare the baseline (RoBERTa-Large with CE loss) against the proposed objective (RoBERTa-Large with CE+SCP loss). There are 4 sets of experiments:
1) When training on the full datasets, results are quite modest (+0.4 increase in accuracy on average over 6 GLUE tasks).
2) In the few-shot setting, CE+SCP does meaningfully better than the baseline (for instance, when fine-tuning on only 20 data points, CE+SCP improves accuracy by more than 10%); these gains decrease as the dataset size increases.
3) When the datasets are noisy (effect obtained via back-translation), CE+SCP shines again (for instance, when the degree of corruption is very high, MNLI accuracy goes from ~47% up to ~53%).
4) Finally, the authors look at domain shift; they fine-tune a model on SST-2, then apply few-shot learning on other sentiment classification datasets. This set of experiments has quite high error margins, so I didn't find it as convincing as 2) and 3).

Here are some questions/suggestions for the authors regarding their experiments:

a) "In all our experiments, [...] we sample half of the original validation set of GLUE benchmark and use it as our test set, and sample ~500 examples for our validation set from the original validation set [...]" -- Evaluating the models on a *subset* of the validation set makes it harder to compare it against other papers that fine-tune RoBERTa-Large. I think that, at least for Table 2, it would be useful for posterity if you could either i) get the true test scores from the GLUE server, or ii) use part of the training set for validation, and then test on the full dev set, which is more standard practice.

b) "We run each experiment with 10 different seeds, and pick the top model out of 10 seeds based on
validation accuracy and report its corresponding test accuracy" -- I am assuming this statement describes how evaluation numbers are reported for a fixed set of hyperparameters. Why do you choose to pick the *top* model as opposed to reporting the *average* accuracy across the 10 runs?

c) "we observe that our proposed method does not lead to improvement on MNLI [...]. We believe
this is due to the fact that number of positive example pairs are quite sparse [...] with batch size 16 [...]. We show evidence for this hypothesis in our ablation studies that we show in Table 3" -- Then why doesn't Table 3 include MNLI? Am I missing something?

d) This method excels in the few-shot setting, at least compared to the CE baseline. So I think it would be a lot more impactful to focus on this particular use case and convince the reader that CE+SCP is better than some other standard few-shot learning baselines (e.g. meta-learning objectives). I do appreciate that the current message of the paper is crystal-clear (adding a SCP term to the loss leads to better fine-tuning), but I also think that the results in Table 2 are too weak for this somewhat general statement. There is quite a bit of real-estate in the paper that could be re-allocated to something more substantive (e.g. Table 1).

Strengths:
- The presentation of the paper is extremely clean, and the goal is clear.
- In the few-shot learning scenario, CE+SCP performs meaningfully better than the CE baseline.

Weaknesses:
- The main weakness is related to my suggestion d) above. I believe marketing CE+SCP as a general fine-tuning solution with somewhat underwhelming results in Table 2 is a missed opportunity to lead with potentially strong results on few-shot learning. I'm calling the results "underwhelming" because there is evidence that a thorough hyperparameter sweep can boost fine-tuning accuracy on GLUE by quite a bit. For instance, Dodge et al. [1] show that fine-tuning BERT carefully can increase SST-2 accuracy by ~2% without any changes in the pre-trained model or fine-tuning objective.

[1] Dodge et al., Fine-Tuning Pretrained Language Models: Weight Initializations, Data Orders, and Early Stopping

---

> ### Author Response · Authors · 2020-11-25
> **Response to Review #2**
>
> Thank you for your helpful feedback and kind remarks on the presentation of our paper and our experimental results on few-shot learning. We revised the paper to include p-values for few-shot learning and full dataset experiments to address concerns about the significance of the results. We respond to individual questions below.
>
> “a) : Sampling of validation set and full dataset evaluation”: We previously sampled the validation set as described in order to be consistent with the few-shot learning results. However, as you and other reviewers rightfully pointed out, this raised concerns about the comparability of our method to the previous methods. We revised our full dataset results (Table 5) to show the test performance on the original GLUE validation set, as presented in the original RoBERTa paper. [3] We sampled training and validation sets from GLUE’s original training set during fine-tuning.
>
> “b): Top vs. average accuracy”: Our statement describes how evaluation numbers are reported for a fixed set of hyperparameters. We revised our paper to report the average test accuracies across 10 seeds instead of top model performance.
>
> “c) Relation between batch size and performance”: In our ablation experiment, our goal is to show that there is a correlation between performance and batch size. We revised our ablation study, Table 6, to include MNLI along with a comparison on training speed measured by average updates per second across different batch sizes.
>
> “d) Few-shot results are strong, while full dataset results are modest.”: We completely agree that SCL loss provides more statistically meaningful improvement in the few-shot learning setting compared to the full dataset setting. We unfortunately did not have enough time during the rebuttal period to compare to the other popular few-shot learning methods. We will investigate that direction in our future work -- thank you for your suggestion!

---

### Official Review · AnonReviewer1 · 2020-10-27
**The results look good but we still need more experiments to identify the source of improvement**

**Rating:** 7
**Confidence:** 3

**Review:**

This work adds a SCL (supervised contrastive learning) loss term during the fine-tuning stage of RoBERTa. The results show that the model with the SCL term and cross-entropy (CE) achieve better GLUE scores than the classic baseline that only uses CE loss, especially when the numbers of supervised training data are small and the data is noisy.

Pros:
The method is simple and the improvement looks significant under various settings

Cons:
It is not very clear whether and why SCL loss improves the results (see the detailed comments below).


Clarity:
The text is fluent and the paper cites lots of related work, but the paper does not well explain why the method performs well.

Originality:
The SCL is not novel because it comes from computer vision but this is the first paper I have seen that successfully applies SCL in NLP tasks.

Significance of this work:
If the authors can really show that the improvement comes from SCL, it may become a popular tool in the fine-tuning stage.


It is possible that the source of improvement comes from the temperature tau and l2 normalization instead of SCL loss itself. Both of the tricks could be also applied to CE loss. Thus, could you do control experiments that replace the SCL loss with the CE loss but keeping l2 normalization and retune the tau and lambda. You can report it as CE+CE.

As shown in Figure 1, the authors suggest that the main reason that SCL loss is better because SCL loss tends to encourage the samples belonging to the same class. However, I believe that CE loss could achieve the same goal and maybe more aggressively than SCL loss. My understanding is that while minimizing CE loss, we encourage each class embedding close to all its samples, so the class embedding tends to become the cluster center of all the points in the class. In the meanwhile, we encourage the samples in each class close to its class embedding, so the samples within the same class will also become closer together, right (e.g., in Figure 3, CE loss could also separate the two classes with a large margin)? The difference is that CE loss encourages the samples close to the average of the samples in the same class, but SCL loss encourages the samples close to each sample in the same class. From this perspective, could you tell us why SCL loss is better (or tell me why this perspective is wrong)? In the introduction, you cite several studies that mention the limitations of cross-entropy loss. I think the motivation of the paper will be much stronger if the authors could briefly and intuitively introduce why CE loss is worse (e.g., what does it mean poor margins and why CE leads to them) and why SCL loss could fix that issue. It would be even better if you can design experiments to demonstrate that (e.g., measure and compare the margins of CE+CE with CE+SCL).

The effect of hyperparameters on performances is not clear. You mention that lambda is tuned in each downstream task. It is possible that in many applications, lambda is 0. Could you show the lambda value for each downstream application? In addition, could you show the final tau value(s) as well? In the experiment setup, you mention that you report the best model out of 10 seeds. Could you also report the average performance? This will tell us more about how stable this SCL method is. In Table 3, could you also report CE only and CE+CE performance?

I believe the issues could be resolved by conducting more controlled experiments and analyses, so I currently vote weak accept. If the concerns are not addressed well during the rebuttal period, I am very likely to change my vote to rejection.

Minor:
1. In equation (2), I think putting CE loss for multi-class classification would be more general.
2. Before the "Relationship to Self-Supervised Contrastive Learning", you mention lower temperature creates harder negatives. The meaning of negative here is unclear. I think you mean harder examples here.

---

> ### Author Response · Authors · 2020-11-25
> **Response to Review #1**
>
> Thank you for your thorough and helpful feedback -- your suggestions helped us investigate our experimental settings further and strengthen our understanding.
>
> You insightfully identified that improvement that comes from SCL over CE might be partly attributed to L2 normalization and the temperature scaling parameter tau. We conducted experiments to test this hypothesis and reported few-shot learning results in Table 10 and full dataset results in Table 8 as CE+CE. We observed that indeed CE+CE outperforms CE on the full data experiments as shown in Table 8.  CE+SCL still outperforms both CE and CE+CE on 4 out of 6 tasks, but only the MRPC and QNLI results are statistically significant. On few-shot learning experiments, CE+SCL outperforms both CE and CE+CE across all tasks and data regimes as shown in Table 10. In fact, CE+CE decreases the performance compared to CE for some settings such as the 100 labeled examples case in Table 10 for MNLI and QNLI. Overall, these experiments show that (i) L2 normalization and temperature scaling both matter more than we previously hypothesized, and (ii) few-shot learning improvements are more statistically significant than the full dataset improvements. We have repositioned the work as a solution for low-data regimes instead of a general fine-tuning solution as other reviewers suggested as well.
>
> You made a comment on the intuitive difference between SCL and CE, stating that CE loss encourages the samples close to the average of the samples in the same class, while SCL loss encourages the samples close to each sample in the same class. We agree that both CE and SCL effectively push embeddings of the examples of the same label closer to each other. However, SCL explicitly pushes embeddings of the examples with the same label close to each other and examples with different labels further away by definition, without CE’s projection step to logits in the end. Hence, it encourages even smaller intra-class distance and larger inter-class distance, as we empirically demonstrate the difference in margins in Figure 3 for few-shot learning settings. Our robustness across augmented noisy training datasets experiments shown in Table 3 are motivated by the previous work that we cite in the introduction that suggested CE loss has poor generalization performance, and we empirically investigate whether SCL loss has improved robustness over CE loss.
>
> Based on your comment on \lambda and \tau, in Section 4.1, we discussed how we conduct hyperparameter search and reported hyperparameters for our best performing models for reproducibility. We revised the paper to report average test performance for the full dataset results in all tables, changed CE loss formula to multi-class classification, and included CE+CE results in our ablation for batch size and training speed in Table 9. As a clarification, by “hard negative”, we mean examples of different labels that are hard to separate. For concerns on the novelty of our paper, we would like to refer you to the novelty section in our general response to reviewers.

---

> > ### Comment · AnonReviewer1 · 2020-11-25
> > **The new results make the paper stronger but the reasons behind why SCL is better than CE could still be improved.**
> >
> > By comparing CE+CE with CE+SCL, the authors confirm that the advantage of SCL in few-shot learning settings, so I will change my vote to acceptance.
> > As for the presentation, I still think the revised version still does not intuitively explain why SCL is better than CE. The authors mentioned that previous work shows that CE loss has poor generalization performances in the paper and above response, but I think why that is the case and why SCL can solve that issue to achieve a better generalization performance should be the core of this paper. In the response, the authors seem to claim that CE cannot separate the classes as well as SCL due to CE’s projection step to logits, but why is the projection bad in few shot settings and not so bad when we have many training samples?
> > If the authors cannot intuitively explain these questions, it might mean that the authors haven't understood the reasons behind the observed improvement well enough, which will limit the impact of this work significantly.

---

### Official Review · AnonReviewer3 · 2020-10-28
**SUPERVISED CONTRASTIVE LEARNING FOR PRE-TRAINED LANGUAGE MODEL FINE-TUNING**

**Rating:** 5
**Confidence:** 5

**Review:**

The paper proposes using a combination of two losses in the fine-tuning stage when using a pre-trained model: the standard CE one, plus a supervised contrastive loss SCL (combined linearly via a \lambda hyperparameter). The supervised contrastive loss uses a normalization summation over the batch examples, with a temperature hyperparameter \theta.

The empirical results cover nicely three scenarios: (a) the impact of adding the SCL loss, in the presence of all the fine-tuning data; (b) the impact of adding the SCL loss in few-shot learning scenarios; and (c) the impact of SCL in the presence of training noise (induced via back-translation through German, using a standard WMT-trained MT model).
The results as presented are encouraging, and support the main hypothesis of the paper, namely that training with the added SCL loss improves performance over all three scenarios mentioned above.

I have a few suggestions that could be seen as minor, and a few observations that are major.
Minor ones:
(i) adding the SCL loss clearly impacts the training speed, yet there is no mention of that (especially as a function of the batch size); in particular, Table 3 would offer the perfect place for mentioning how the training time (for a fixed number of training steps) is affected by the increase batch size, so that the reader can understand both the “upsize” (the improved performance)  as well as the “downsize” (training cost).
(ii) I could see no mention of the settings for the hyperparameters used (\lambda and \theta), nor any ablation experiments that would indicate how their values have been chosen; in the interest of both reproducibility and increased understanding of the value of SCL, please add a sub-section that discusses this issue.

Major ones:
(i) It appears that the authors have used the Roberta_large model and run their own experiments with fine-tuning, w/o and w/ the SCL loss, with little regard for reporting against the published numbers for the GLUE tasks; for instance, in Table 2, they report with CE-only to have RTE performance at 85.0, while the Roberta paper shows 86.6 for it; in this case, the CE+SCL at 85.6 no longer looks like a convincing win; a bit different but nevertheless troublesome is the result reported for CoLA, at 86.4 (CE-only); the numbers for CoLA are normally much lower than that, eg the Roberta paper shows 68.0 (CE-only); this disparity throws a lot of doubt over the accuracy of the results reported in Table 2.
(ii) I commend the authors for showing the variance across their results in both Table 3, 4, and 5; however, it is unclear to me that the claims that the CE+SCL approach is better are being supported by the results. It is not like the CE+SCL gives lower variance, as it is clearly the case that sometimes it is higher than the CE alone; and the fact that, under this high variance, the CE method sometimes performs better CE+SCL puts the whole conclusion under doubt from an empirical standpoint.

One one result that seems to hold strong is the one for few-shot learning, which appears to support the main hypothesis of the paper. However, the main issues mentioned above would need to be addressed in order to have the paper reach the level of clearing the bar for ICLR publication.


Re: References A lot of the references use the Arxiv version for papers that have been peer-reviewed and published. Please fix.

---

> ### Author Response · Authors · 2020-11-25
> **Response to Review #3**
>
> Thank you for your positive comments on our few-shot learning results and your insightful feedback.
>
> Based on your suggestions, we updated our full dataset results in Table 5 such that they show the test performance on the original GLUE validation set, as presented in the original RoBERTa paper [3]. There might still be discrepancies between the original RoBERTa paper results and ours since (i) RoBERTa paper reports the median of 5 seeds while we report the average of 10 seeds, (ii) training and validation dataset samples are different, (iii) small datasets such as RTE, CoLA, and MRPC have inherently high variance during fine-tuning [4], and (iv) CoLA results are not comparable as RoBERTa paper reports Matthews correlation coefficient as their performance metric while we report accuracy.  Also, we included p-values for our few-shot learning experiments in Table 2 and for full dataset experiments in Table 5 to show the statistical significance of our results, and finally in Section 4.1 we discussed how we conduct hyperparameter search and reported hyperparameters for our best performing models for reproducibility.
>
> We would like to clarify that supervised contrastive loss calculation is negligible given the rest of the model computation, when we are using a batch size of 16 as we do in all of our experiments with RoBERTa_Large due to memory constraints. To support this claim, we added a comparison to Table 6 that shows training speed measured by average updates per second across different batch sizes. Also, we fixed the references that use the arXiv version of published papers -- thanks for pointing that out!

---

### Official Review · AnonReviewer4 · 2020-10-28
**Good paper but small contribution**

**Rating:** 6
**Confidence:** 4

**Review:**

Summary
* For the fine-tuning of pre-trained language models, the authors proposed a supervised learning method that combines cross-entropy loss and contrastive loss.
Experimental results show that the proposed method improves over cross-entropy loss on several classification tasks of the GLUE benchmark set.
The proposed method outperforms cross-entropy loss in few-shot learning tasks and noisy datasets generated by English-German and German-English translation.

Strong points

* The proposed loss function is reasonable and the effect of supervised contrastive learning was not reported for NLP applications before, the experimental results are valuable.
* The paper is well-organized and well-written.
* Without using extra datasets for fine-tuning, the proposed method consistently improves the baseline method.
* The generation of noisy examples using the back-translation model in Section 5.3 is an interesting approach to analyze model robustness.

Weak points
* Although the supervised contrastive learning term as previously proposed in (Khosla et al. 2020), it is not cited in the section.
* The benchmark results in Table 2 are not comparable with conventional methods since the experimental setting does not follow the finetuning procedures from prior work (Devlin et al., 2019) which reports the test set performance obtained from GLUE submissions.

Decision reason
* The technical contribution of this paper is limited since the proposed method is a rather strait-forward expansion of Khosla et al. 2020.  In addition, although it is novel to apply supervised contrastive learning for NLP applications, the impact of these results is also limited because the experimental results are not directly comparable with previous work.

Questions
* Why did you "sample ... taking the label distribution of the original validation set into account" in Section 4.1?   I am worried that this sampling procedure may make the few-shot task easier.

Additional Feedback
* Since the subtraction between two values in percentage is not a ratio,    the percentage is not an appropriate unit for it.  For example "1.2% improvement on SST-2" in Section 5.1 should be "1.2 point improvement on SST-2".
* Since Khosla et al. 2020 proposed a two-stage training procedure, supervised contrastive learning at the first stage and the learning of the output layer at the second stage, I would like to see the qualitative comparison with the proposed joint training procedure.
* Instead of using different original sentences for each T values, it is clear and compact to use the same original sentence for each T values.

---

> ### Author Response · Authors · 2020-11-25
> **Response to Review #4**
>
> Thank you for your helpful feedback and kind remarks on our experimental results and the organization of our paper.
>
> You rightfully point out that the supervised contrastive loss was previously proposed by Khosla et al. (2020), although our work is the first to apply to natural language processing tasks. We consider Khosla et al. (2020) as concurrent work, and we cite them in our related work. Based on your suggestion, we include a comparison with their two-stage training procedure for our full dataset experiments in Table 8 in the Appendix. We also would like to emphasize that our paper’s main focus is performance and robustness in the few-shot learning settings, unlike previous work where they have access to a lot of labeled data.
>
> Also, we updated our full dataset results in Table 5 such that they show the test performance on the original GLUE validation set, as presented in the original RoBERTa paper. [3] This will hopefully make our work more directly comparable with the previous methods. We address your individual questions below:
>
> “Instead of using different original sentences for each T value, it is clear and compact to use the same original sentence for each T value.”: We use the same original sentences for each T value for our robustness experiments presented in Table 3. We include different original sentences in Table 4 to show different qualitative examples.
>
> “Why did you "sample ... taking the label distribution of the original validation set into account" in Section 4.1? I am worried that this sampling procedure may make the few-shot task easier.”: We sampled taking the label distribution into account in order to keep the data distributions consistent with the original train and validation datasets. We would appreciate any additional insight on why you think this sampling procedure might make the few-shot task easier.

---

### Author Response · Authors · 2020-11-25
**General Response to Reviewers**

We thank all the reviewers for their valuable time and insightful feedback. We are encouraged that they find our results significant (R1), encouraging (R3), and valuable (R4), particularly in the few-shot learning scenario (R2, R3) and robustness across augmented noisy training set scenario (R2); think it has the potential to become a popular tool in the fine-tuning stage of pre-trained language models (R1); find our proposed way of analyzing model robustness interesting (R4); and overall find the paper well-written (R1, R2, R4). Based on the feedback of all reviewers, we have repositioned our work as a solution for low-data regimes instead of a general fine-tuning solution. We address the concerns that are shared across several reviewers below.

Significance of Results: R2, R3, and R4 all requested to see full dataset results that are more easily comparable to previous work. We updated our full dataset results shown in Table 5 such that they show the test performance on the original GLUE validation set, as presented in the original RoBERTa paper. [3] We sampled training and validation sets from GLUE’s original training set during fine-tuning. Also, R2 and R3 had concerns about the statistical significance of our experiments -- we revised our paper to include p-values for full dataset results in Table 5, and for few-shot learning results in Table 2.

Novelty: R1 and R4 expressed concerns about the novelty of our work stating that supervised contrastive learning comes from computer vision. We consider Khosla et al. (2020) [1] and Liu et al. (2020) [2] concurrent work, and we cite both of them in our paper in related work. We also included a comparison with the method proposed in Khosla et al. (2020) based on the request of R4 in Table 8 in the Appendix for the full dataset results. As pointed out by both R1 and R4, we would like to emphasize that our work is the first successful application of supervised contrastive learning in the context of natural language processing to the best of our knowledge. In addition, unlike previous work in computer vision, our paper’s main focus is few-shot learning, along with robustness across augmented noisy training sets when we have few labeled examples.

We addressed reviewers’ individual comments including hyperparameter specifications (R1, R3), citations (R3), training speed (R3), and few other specific questions.

[1] Prannay Khosla, Piotr Teterwak, Chen Wang, Aaron Sarna, Yonglong Tian, Phillip Isola, Aaron Maschinot, Ce Liu, and Dilip Krishnan. Supervised contrastive learning. In NeurIPS, 2020.

[2] Hao Liu and P. Abbeel. Hybrid discriminative-generative training via contrastive learning. ArXiv, abs/2007.09070, 2020.

[3] Y. Liu, Myle Ott, Naman Goyal, Jingfei Du, Mandar Joshi, Danqi Chen, Omer Levy, M. Lewis,Luke Zettlemoyer, and Veselin Stoyanov. Roberta: A robustly optimized bert pretraining approach. ArXiv, abs/1907.11692, 2019.

[4] Tianyi Zhang, Felix Wu, Arzoo Katiyar, Kilian Q. Weinberger, and Yoav Artzi. Revisiting few-sample bert fine-tuning. ArXiv, abs/2006.05987, 2020.

---

### Decision · Program_Chairs · 2021-01-07
**Final Decision**

**Decision:**

Accept (Poster)

**Comment:**

This paper introduces supervised contrastive learning loss on top of typical cross-entropy loss for fine-tuning language model for downstream tasks. While the idea is simple and has been used in vision literature (as pointed out by R1 & R4), its application LM is first introduced in this paper. The experimental gain is small in the regular setting but clearer gains in a few-shot learning setting and noisy training dataset (through back translation) setting. Overall the paper is clearly written and experiments are carefully studied. During the discussion phase, the authors provided results on the full GLUE dataset as well as other ablation studies (e.g., CE+CE recommended by R2), improving the paper.